# Community Pharmacy Minor Ailment Service (PMAS): An Untapped Resource for Children and Their Carers

**DOI:** 10.3390/pharmacy9020102

**Published:** 2021-05-17

**Authors:** Tami Benzaken, Godwin Oligbu, Michael Levitan, Subrina Ramdarshan, Mitch Blair

**Affiliations:** 1Department of Paediatrics, North West London University Healthcare Trust, Harrow HA1 3UJ, UK; tami.benzaken@nhs.net; 2Department of Paediatrics, St George’s University of London, London SW17 0QT, UK; godwin.oligbu@nhs.net; 3Middlesex Group of Local Pharmaceutical Committees, London N20 9HH, UK; Michael@middlesexpharmacy.org (M.L.); subrina@middlesexpharmacy.org (S.R.); 4Department of Primary Care and Public Health, Imperial College London, Harrow HA1 3UJ, UK

**Keywords:** minor illness, community pharmacy services, primary healthcare, minor ailment service, child, policy

## Abstract

Background: The Pharmacy Minor Ailment Service (PMAS) was introduced in the UK over 15 years ago for use in treating minor ailments and has been shown to be effective and acceptable by the public in reducing the burden on high-cost healthcare settings (such as general practice and emergency departments). This paper aims to review the use of a PMAS in the paediatric population. Methods: PMAS was established in a London Borough in 2013. Data were collected from 33 pharmacists and 38 GPs on demographics, service utilization and costs. Results: In total, 6974 face-to-face consultations by 4174 patients were provided by pharmacies as part of the PMAS over a 12-month period. Moreover, 57% of patients were children with fever, hay fever and sore throat, accounting for 58% of consultations. Only 2% were signposted to other services. Sixty-nine percent of patients reported being seen within 5 min and 96% of patients were seen within 10 min with high levels of satisfaction. Cost savings of over GBP 192,000 were made during the scheme. Conclusions: PMAS is a highly cost effective, accessible and acceptable service for children with minor illnesses.

## 1. Introduction

Minor ailments are defined as “common or self-limiting, uncomplicated conditions that can be diagnosed and managed without medical intervention” [1,2]. In the UK, it is estimated that between 18% and 37% of 57 million annual GP consultations are for minor ailments [3,4,5]; costing an estimated GBP 2 billion per annum [6]. Additionally, 5% of consultations in Emergency Departments (ED) could be managed in community pharmacies instead [7].

Hendry et al. found that 79% of the paediatric ED attendances were for minor injuries and illnesses and these children were more likely to fall into a younger age group [8]. Approximately 10% of infants attending an ED in a large district general hospital had no coded diagnosis and probably attended for minor illness and general advice only [9]. As this is a growing proportion of ED attendances, this puts a lot of demand on our health services [10]; greater use of community pharmacy services instead of medical services for minor ailments could help relieve pressure on healthcare providers in high-cost settings [11,12].

The ‘NHS Plan’ was published in July 2000 and outlined a number of changes and improvements to the National Health Service (NHS) [13]. A key theme was ‘patient-centred’ care. The NHS Plan made several recommendations on how pharmacies could integrate better within the NHS for the benefit of patients. These ideas were reinforced in the strategic document ‘Pharmacy in the future’ [14] and as a consequence, the new pharmacy contractual framework was established in 2005 [15]. These schemes were introduced nationally in all community pharmacies in Scotland and Northern Ireland in 2006 and 2009, respectively. This was also rolled out in Wales in 2013 [16].

More recently, in January 2019, the NHS published the Long Term Plan (LTP) which stated the Government would make greater use of community pharmacists’ skills and opportunities to engage patients [17]. They highlighted a clear plan to help people live well and age well, with the aim of improving out-of-hospital care by supporting health professionals in the community provide a better service. A minor ailments service accessible to everyone from all community pharmacies would be an important and effective way to achieve these aims.

In England, Pharmacy Minor Ailment Schemes (PMAS) are specified as ‘enhanced’ services within the community pharmacy contract, which can be commissioned by the Clinical Commissioning Groups (CCGs) after the assessment of local needs.

In a recent retrospective review by Fielding et al. [18] in North East Scotland where minor ailments suitable for management in community pharmacies were categorised, it was estimated that 18 million general practice and 6.5 million ED consultations could be redirected to community pharmacy, equating to approximately GBP 1.1 billion saved in resources per year [19]. Similar findings were observed in the North East of England where the PMAS was estimated to reduce local healthcare costs by GBP 6739 per month [20].

The MINA study (a programme which derived evidence to inform recommendations regarding the future delivery of community pharmacy-based minor ailments schemes in the UK) corroborated this: 13% and 5% of consultations in general practice and ED, respectively, are for minor ailments that could be managed in community pharmacies [7]. 

One of the pertinent issues raised in most studies was patient satisfaction. The systematic review conducted by Paudyal et al. [21] noted that the proportion of patients reporting complete resolution of symptoms after an index PMAS consultation ranged from 68% to 94%, and that re-consultation rates in general practice, following an index consultation with a PMAS, ranged from 2.4% to 23.4%. 

It is therefore unclear why community pharmacy services are not used more often for the management of minor ailments. Previous research investigating the use of community pharmacies [22] and preferences for managing symptoms of minor ailments [23] suggests that factors such as age, gender, nature of symptoms and previous experience influence community pharmacy use. Different strategies may therefore be required to reduce the demand on general practices and EDs and ensure patients with minor ailments can access care appropriately from the community pharmacy.

While these studies tell us something about the service attributes, none, to our knowledge, have specifically estimated the use of PMAS in children. In this study, we look at PMAS utilisation by children below the age of 16, parents’ satisfaction and re-consultation rates by age group. 

## 2. Materials and Methods

In February 2013, a Minor Ailment Service (MAS) was launched in the London Borough of Enfield commissioned by the local CCG as a service level agreement. 

The service was offered to all pharmacies and was rolled out in waves. This consisted of a total of 33 pharmacies. The pharmacies had patients referred from 38 participating GP practices. All GP practices in Enfield were offered to participate in the scheme. 

The pharmacies were provided with minor ailment protocols and a joint medication formulary. Participating pharmacists had to complete and pass Centre for Pharmacy Postgraduate Education (CPPE) training on minor ailments and attend a training event on the service held by NHS Enfield before they were able to deliver the service. In total, the pharmacies were able to provide advice for 20 minor ailments with a choice of 76 over the counter medications as treatment options. 

The training event was facilitated by NHS Enfield CCG. Sessions covered how the PMAS worked, the formulary used, patient criteria, how the IT system worked, case scenarios and a question-and-answer session. 

All consultations over a 1-year period were recorded and data collected (between 18/2/13–28/2/14). 

Data were collected on patient’s age, presenting ailment, time to consultation and patient feedback of the service. The data were collected using the North 21 electronic platform and the data were analysed by NHS Enfield Primary Care Team and Medicine Management.

## 3. Results

A total of 6974 face-to-face consultations by 4174 patients were provided by pharmacies as part of the PMAS over a 12-month period. Of these, 85% (3535) of patients were seen on no more than two occasions.

The service was most highly utilised by the paediatric population with over half of consultations for those 16 years and under (57%, *n* = 4001). Of these consultations, over half (52.9%, *n* = 2119) were for patients aged 5 and younger. 

Only 143 patients (2%) utilising the service required signposting to other healthcare providers, and of these, for 96% (*n* = 138), the source of referral was back to the GP. 

The three most common ailments presented to the PMAS were fever, hay fever and sore throat, accounting for 58% (*n* = 4103) of the total activity (see Figure 1). 

A total of 4923 patients provided feedback on the PMAS, accounting for 70% of the entire activity provided. Sixty-nine percent of patients reported being seen within 5 minutes and 96% of patients were seen within 10 minutes. 

When questioned where would the patient have gone had they not used the PMAS, 90% (*n* = 3866) stated they would have gone to the GP. A further 4% (*n* = 166) said they would have gone to Accident and Emergency and 3% (*n* = 125) would have gone to a walk-in-centre. 

The cost per consultation using the PMAS was estimated to be GBP 11.56, inclusive of start-up plus non-recurrent costs, or GBP 7.37, excluding start-up costs. The service consultation costs were modelled on pre-existing PMAS identified nationally at the time. These costs accounted for the level of input required by pharmacists to provide consultations and supply of medicine. This was used to compare the average costs of utilising other services at the time in order to calculate the potential savings resulting from this service. An estimated gross savings of GBP 192,598.58 was calculated based on patient feedback (see Table 1 for a breakdown of the calculated costs), with a net saving of GBP 111,612.07. 

Individual pharmacies saw an increase in activity following the introduction of the PMAS, this was particularly evident in pharmacies located in more deprived areas. The demand for consultations in pharmacies was additionally demonstrated as NHS Enfield had to implement a capping system for the number of consultations per pharmacy. 

Pharmacists were able to make positive contributions to the standardised formulary. Following the most common conditions seen, they suggested including the use of chloramphenicol for minor eye infections, introducing Patient Group Directions (PGDs) to allow the supply of antibiotics for urinary tract infections and allowing the supply of simple linctus for coughs due to its placebo effects. 

## 4. Discussion

This is the first population-based systematic study of the use of community pharmacy services for parents and children for minor ailments. It demonstrated that pharmaceutical services have the means to provide an effective and efficient service to patients with minor ailments, reducing pressures on other healthcare services (particularly GP services). 

The service demonstrated that pharmacies are able to provide quicker services to patients with much shorter waiting times compared to other healthcare services used. The majority of patients using the PMAS were seen within 5 minutes and almost all (96%) were seen within 10 minutes. This is in contrast to the ongoing long waiting times for GP appointments. In 2018, almost a third of patients in the UK had to wait more than a week for a GP appointment (32.8%) [24]. The average waiting time in Accident and Emergency for patients to be seen is 2 h and 28 min [25]. This is substantially longer than the waiting time for almost all patients utilising this service. 

The PMAS appears to have had good clinical outcomes and almost all of the patients seen were able to be treated satisfactorily within this setting. Only 2% required onwards referral to other healthcare providers, predominantly the GP (96%). This, coupled with the fact that service users rated the service highly in terms of satisfaction levels, demonstrates the effectiveness of the PMAS. Additionally, the data show that commissioned minor ailments services deliver an estimated net value per intervention of GBP 59.08, thus delivering a higher quality service, compared with the cost of the intervention [26]. However, it is impossible to determine from the data how many of the patients seen went on to seek further medical advice from other healthcare providers at a later date. 

A number of limitations were encountered as part of the PMAS. Primarily, these related to the way patients were able to access the service. Patients who wanted to utilise the PMAS required Patient Passports. These had to be issued by the GP through the CCG. This added a further stage of complexity in accessing PMAS for patients. Alternative methods of delivering the Patient Passports may have allowed for smoother access to the service. Additional limitations of the PMAS included the limited number of conditions in the formulary as well as financial constraints, which limited the scope of the service. 

Included in the LTP is the development of Primary Care Networks (PCNs), which are a group of local health professionals responding to the needs of the patients they serve. This plays a key part in the roll out of the LTP, which includes community pharmacists. They are already working alongside GPs and other primary care teams and ensure millions of patients safely receive the medicines they need when they need them. The PMAS (which has now transitioned to the Community Pharmacy Consultation Service) would allow community pharmacists to contribute to the LTP to a greater degree and enable further integration into the primary healthcare team, benefiting everyone who visits their local pharmacy on a regular basis.

Community pharmacies continue to provide many essential services to women and children. In 2020, around 1 in 20 pregnant women who received flu immunisation had this given by a pharmacist and of course many are currently supporting the roll out of the COVID-19 vaccine. This could be extended to high-risk women and children as part of an integrated primary care service, giving families a further option other than their GP or hospital teams. 

Further work is required to look into the possibility of developing a second-tier pharmacy service to develop accredited ‘Paediatric Practitioner Pharmacists’ working in community pharmacies from PGDs as well as accredited ‘Advanced Paediatric Practitioner Pharmacists’ with further physical assessment skills and independent prescribing qualifications. The need for such roles has been highlighted in the Analysis of Minor Ailment Services Data (July 2017), which showed 61% of consultations were for people aged under 16 years [27]. 

Pharmacies are an untapped resource with the potential to provide large savings to the NHS, both financial savings and time savings. The average costs of utilising other healthcare providers are all greater than the cost per patient of using the PMAS (between approximately 2–6 times the cost of the PMAS). The project estimated total savings of almost GBP 200,000; this is across one London borough over 1 year. If the PMAS were to be expanded over all London boroughs, the projected financial savings would be over GBP 6 million: massive potential savings to already overstretched NHS services. Furthermore, in the shadow of the ongoing COVID-19 pandemic, we believe that many people would prefer to access local community services as opposed to attending hospitals for non-urgent illnesses.

## Figures and Tables

**Figure 1 pharmacy-09-00102-f001:**
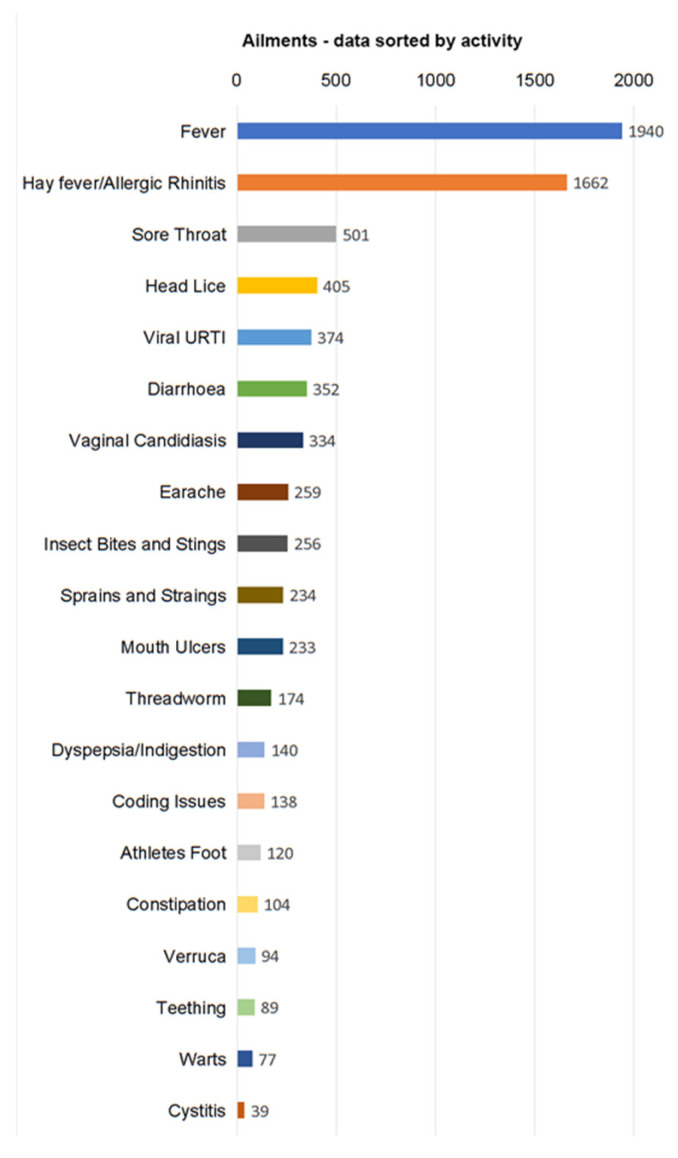
Number and type of ailments seen in the 12-month period.

**Table 1 pharmacy-09-00102-t001:** Summary of Savings.

Description	GP Practice	A&E Setting *	Walk in Centre	Urgent Care Centre	Bought a Medicine	Phoned 111 for Advice	Used Out Of Hours Doctors Service
**Tariff Costs per Slot**	£45	£58	£63	£58	N/A	Unknown	£16.60
**Total Appointments**	3866	166	125	19	103	7	7
**Total Cost**	£173,970	£9628	£7875	£1102	£92.62	Unknown	£116.20
**Total**							£192,598.58

* A&E Setting: Patients seen within the Emergency Department.

## Data Availability

No access to original data is provided.

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
