# Peer review of "Community Pharmacy Minor Ailment Service (PMAS): An Untapped Resource for Children and Their Carers"

_pharmacy, 2021, doi:10.3390/pharmacy9020102_

Round 1

Reviewer 1 Report

The paper covers an important topic and the study is innovative. I only have two suggestions for the authors' considertation:

  1. Cost-of-service calculation for PMAS could be more detailed. The reported estimate seems low to me. Was the complete cost-of-service considered including costs such as space/rent, utilities, administrative, billing, equipment, software, training? It seems like the reported cost might only cover the pharmacists' time, but cost-of-service would certainly be more than that.
  2. Readers might be interested in work systems that were set up and processes of care that were employed for service delivery. Were there challenges at various sites? Were there success stories? Some examples include computers, software, patient interaction space, privacy, equipment, communications, personnel management, personnel training, approvals, record-keeping, patient follow-up, billing, etc.

Author Response

Many thanks for your comments. Please see responses below to individual points you made.

1. Cost-of-service calculation for PMAS could be more detailed. The reported estimate seems low to me. Was the complete cost-of-service considered including costs such as space/rent, utilities, administrative, billing, equipment, software, training? It seems like the reported cost might only cover the pharmacists' time, but cost-of-service would certainly be more than that.

Response: Many thanks for your comments. Project management was provided internally by the CCG (clinical Commissioning Group) at a neutral cost. This included a project manager, GP Clinical Lead, Medicine Management lead and LPC (Local Pharmaceutical Committee) lead. The IT software cost is commercially sensitive. The costs for space and training were included in the price for the consultation. I hope this answers your query.

2. Readers might be interested in work systems that were set up and processes of care that were employed for service delivery. Were there challenges at various sites? Were there success stories? Some examples include computers, software, patient interaction space, privacy, equipment, communications, personnel management, personnel training, approvals, record-keeping, patient follow-up, billing, etc.

Response: Whilst we did not have the scope to gather individualised qualitative feedback from pharmacies, we do have some anecdotal success stories and challenges which were observed. I have included the following paragraph on success stories seen across individual pharmacies in increased activity on page 5, below table 1 (tracked):

‘Individual pharmacies saw an increase in activity following the introduction of the PMAS, this was particularly evident in pharmacies located in more deprived areas. The demand for consultations in pharmacies was additionally demonstrated as NHS Enfield had to implement a capping system for number of consultations per pharmacy.’

I have also now included a paragraph detailing limitations of the study, which includes challenges encountered. This can be found as paragraph 4 in the discussion section. The paragraph reads as follows:

‘A number of limitations were encountered as part of the PMAS. Primarily these related to the way patients were able to access the service. Patients who wanted to utilise the PMAS required Patient Passports. These had to be issued by the GP through the CCG. This added a further stage of complexity in accessing PMAS for patients. Alternative methods of delivering the Patient Passports may have allowed for smoother access to the service. Additional limitations of the PMAS included the limited number of conditions in the formulary as well as financial constraints, which limited the scope of the service. ‘

Reviewer 2 Report

This is an interesting manuscript. However, English language should be improved throughout the manuscript. Furthermore, I have few comments:

  1. introduction line 65 please provide what MINA abbreviation stands for, and add few sentences about this study
  2. methods line 88 what is CCG, line 91 write the exact number of all GP practices
  3. methods line 96 please add full description of training event, e.g. what lectures were included
  4. results line 121 it is not usual that sentence starts with a number
  5. please provide study limitations at the end of discussion section

Author Response

Many thanks for your comments. Please see our responses below.

1. Introduction line 65 please provide what MINA abbreviation stands for, and add few sentences about this study.

Response: Many thanks for highlighting this. MINA does not appear to be an abbreviation, but simply the name given by the authors to the study. However, I have now added a few sentences about the study. I have now amended this line (tracked) accordingly and it now reads as follows:

‘The MINA study (a programme which derived evidence to inform recommendations regarding the future delivery of community pharmacy-based minor ailments schemes in the UK) corroborated this; 13% and 5% of consultations in general practice and ED respectively are for minor ailments that could be managed in community pharmacies [7].' 

2. Methods line 88 what is CCG, line 91 write the exact number of all GP practices.

Response: CCG refers to Clinical Commissioning Group. Please see line 58 of introduction where this is written out in long hand. Line 91: Unfortunately, we do not have the total number of GP practices in Enfield from 2013 as the configuration of Primary Care in the UK has undergone significant re-structuring in the last few years, which means that the current total number of practices is significantly different to 2013.

3. Methods line 96 please add full description of training event, e.g. what lectures were included.

Response: I have added the following details of the training events (tracked):

‘The training event was facilitated by NHS Enfield CCG. Sessions covered how the PMAS worked, the formulary used, patient criteria, how the IT system worked, case scenarios and a question and answer session.’

4. Results line 121 it is not usual that sentence starts with a number.

Response: Many thanks for highlighting this. The sentence has been changed and now reads as follows:

‘A total of 4923 patients provided feedback on the PMAS, accounting for 70% of the entire activity provided.’

5. please provide study limitations at the end of discussion section.

Response: Many thanks for highlighting this. I have now included the following paragraph around project limitations in the discussion section (now paragraph 4 in the section) after discussion of the study results, but prior to the wider discussion.

‘A number of limitations were encountered as part of the PMAS. Primarily these related to the way patients were able to access the service. Patients who wanted to utilise the PMAS required Patient Passports. These had to be issued by the GP through the CCG. This added a further stage of complexity in accessing PMAS for patients. Alternative methods of delivering the Patient Passports may have allowed for smoother access to the service. Additional limitations of the PMAS included the limited number of conditions in the formulary as well as financial constraints, which limited the scope of the service.’

Reviewer 3 Report

The empowerment of community pharmacists and their increased participation in patient’s treatment is a paramount issue.

In this line, this manuscript contributes to demonstrate the benefits of giving them more competencies.

Some things I would like to address:

  • Material and methods. The manuscript is based in a work done during 2013-14. This is some time ago, why did the author not published these results before? Could the results found be extrapolate to the present?
  • Results:
    • How the estimated cost per consultation is calculated?
    • Due to the importance of the financial savings, I think that the authors should explain a little more this point
    • Table 1. Indicate the meaning of A&E Setting in the table footer.
    • It would also interesting, to know some examples of the recommendations given by the pharmacists for some of the consultations. At least for the more frequent.

Author Response

1. Material and methods. The manuscript is based in a work done during 2013-14. This is some time ago, why did the author not published these results before? Could the results found be extrapolate to the present?

Response: Many thanks for your comment. We agree that we would have liked the work published sooner. However, unfortunately, the clinician who had been writing the work up a few years ago was unable to complete the work due to other commitments. We felt strongly that the work done was important in both the empowerment of community pharmacists as well as optimising patient care. Hence, we were keen to persevere and publish the work done despite the delay. We believe the results found can be directly extrapolated to the present. In fact, the actual value and cost savings is likely to be very similar to current times. £1 in 2014 is equivalent to £1.1 in 2021.

2. Results. How the estimated cost per consultation is calculated? Due to the importance of the financial savings, I think the authors should explain a little more on this point.

‘The cost per consultation using the PMAS was estimated to be £11.56 inclusive of start-up plus non-recurrent costs, or £7.37 excluding start-up costs. The service consultation costs were modelled on pre-existing PMAS identified nationally at the time. These costs accounted for the level of input required by pharmacists to provide consultations and supply of medicine. This was used to compare to average costs at the time of utilising other services to calculate potential savings utilising this service. An estimated gross savings of £192,598.58 was calculated based on patient feedback (see Table 1 for breakdown of costs calculated), with a net saving of £111,612.07.’

3. Table 1. Indicate the meaning of A&E Setting in the table footer.

Response: This has been clarified in table footer and the following explanation added: *A&E Setting: Patients seen within the Emergency Department.

4. It would also interesting, to know some examples of the recommendations given by the pharmacists for some of the consultations. At least for the more frequent.

Response: Many thanks for highlighting this. I have added a paragraph on some of the more common recommendations made by pharmacists in order to improve the formulary. These are included now as the last paragraph in the results section, which reads as follows:

‘Pharmacists were able to make positive contributions to the standardized formulary. Following the most common conditions seen they suggested to: include the use of chloramphenicol for minor eye infections, introduce Patient Group Directions (PGDs) to allow the supply of antibiotics for urinary tract infections and to allow the supply of simple linctus for coughs and its placebo effects.’